# Learnable Commutative Monoids for Graph Neural Networks

**Euan Ong**
University of Cambridge
`elyro2@cam.ac.uk`

**Petar Veličković**
DeepMind / University of Cambridge
`petarv@deepmind.com`

## Abstract

Graph neural networks (GNNs) have been shown to be highly sensitive to the choice of aggregation function. While summing over a node's neighbours can approximate any permutation-invariant function over discrete inputs, Cohen-Karlik et al. [2020] proved there are set-aggregation problems for which summing cannot generalise to unbounded inputs, proposing recurrent neural networks regularised towards permutation-invariance as a more expressive aggregator. We show that these results carry over to the graph domain: GNNs equipped with recurrent aggregators are competitive with state-of-the-art permutation-invariant aggregators, on both synthetic benchmarks and real-world problems. However, despite the benefits of recurrent aggregators, their $O(V)$ depth makes them both difficult to parallelise and harder to train on large graphs. Inspired by the observation that a well-behaved aggregator for a GNN is a commutative monoid over its latent space, we propose a framework for constructing learnable, commutative, associative binary operators. And with this, we construct an aggregator of $O(\log V)$ depth, yielding exponential improvements for both parallelism and dependency length while achieving performance competitive with recurrent aggregators. Based on our empirical observations, our proposed *learnable commutative monoid* (LCM) aggregator represents a favourable tradeoff between efficient and expressive aggregators.

## 1 Introduction

When dealing with irregularly structured data [Bronstein et al., 2021], neural networks typically need to process data of arbitrary sizes. In such scenarios, the heart of the network is arguably its *aggregation function*—a function that reduces a collection of neighbour feature vectors into a single vector. Indeed, graph neural networks (GNNs) have been shown empirically to be highly sensitive to the choice of aggregator [Veličković et al., 2019, Richter and Wattenhofer, 2020], with a wide range of aggregators (e.g. sum, max and mean) and their combinations [Corso et al., 2020] in common use.

In this paper, we offer a new perspective for studying aggregators, with clear theoretical and practical implications. It can be said that the *true* objective of choosing an aggregator is to make it as simple as possible (i.e. to minimise the sample complexity required) for the parameters of the GNNs to exploit that aggregator in a way that makes it easier to solve the learning problem. Specifically, we study this in the context of learning to *align* the GNN's aggregator to a desirable target aggregation function (as defined in [Xu et al., 2019a]). It is already a known fact that higher alignment implies reduced sample complexity [Xu et al., 2019a], and in the context of algorithmic reasoning, it is well-known that a neural network will be better at learning to imitate an algorithm if its aggregator matches that of the algorithm it is trying to imitate [Veličković et al., 2019, Xu et al., 2020].

However, beyond the realm of learning a task with a concrete aggregator, many real-world problems offer more challenging settings, wherein the optimal aggregator to learn is not clear—but unlikely to be a trivial fixed aggregator. To formalise this notion, while preserving the useful assumption of permutation invariance, we leverage *commutative monoids* as a formalism for both the aggregators supported by GNNs and the (potentially unknown) target aggregators one would wish to align to.

E. Ong and P. Veličković, Learnable Commutative Monoids for Graph Neural Networks. *Proceedings of the First Learning on Graphs Conference (LoG 2022)*, PMLR 198, Virtual Event, December 9–12, 2022.

This formalism allows us to derive several relevant results, including the fact that using any *fixed* commutative monoid $F$ (e.g. sum or max) as an aggregator would compel the GNN to learn a *commutative monoid homomorphism* from $F$ to the target commutative monoid, purely from data. We hypothesise that this is often difficult to do robustly, and verify our hypothesis by demonstrating several instances (both synthetic and real-world) where fixed aggregators (including combinations of them [Corso et al., 2020]) fail to generalise.

Our perspective, inspired by the functional programming motif of *folds* (or *catamorphisms*) over arbitrary data structures, leads us to consider flexible and learnable aggregation functions, which can more easily fit a wide range of commutative monoids directly, without needing to learn such a homomorphism. The most popular such aggregator has previously been the RNN (i.e. 'a fold over a list') – used, for instance, in GraphSAGE [Hamilton et al., 2017]. The reason for RNNs' expressive power is simple: their usage of a *hidden recurrent state* allows them to break away from the constraints of commutative monoids and aggregate inputs more flexibly. However, while empirically powerful, the sequential structure of RNN aggregators leads to clear shortcomings in efficiency: if an RNN had learnt to aggregate $n$ neighbours under a commutative monoid operation $\oplus$, it would do so with a depth that is linear in $n$, as $\big((((\ldots(\mathbf{x}_1 \oplus \mathbf{x}_2) \oplus \mathbf{x}_3) \oplus \ldots) \oplus \mathbf{x}_{n-1}) \oplus \mathbf{x}_n\big)$.

But, by folding over a *binary tree* instead of a *list* (in other words, rearranging the order of operations to a balanced binary tree $(\ldots((\mathbf{x}_1 \oplus \mathbf{x}_2) \oplus (\mathbf{x}_3 \oplus \mathbf{x}_4)) \oplus \cdots \oplus (\mathbf{x}_{n-1} \oplus \mathbf{x}_n) \ldots))$, we derive an aggregator that achieves a favourable trade-off between flexibility and efficiency, empirically retaining most of the performance of RNNs while having a depth that is logarithmic in $n$. We also demonstrate how such layers can be effectively constrained and regularised to respect the commutative monoid axioms (essentially creating a *learnable commutative monoid*), leading to further gains in robustness.

## 2    Motivation

Before exploring GNN aggregators, we first review the structure of a GNN. For a graph $G = (V, E)$ whose nodes $u$ have one-hop neighbourhoods $\mathcal{N}_u = \{v \in V \mid (v, u) \in E\}$ and features $\mathbf{x}_u$, a message-passing GNN over $G$ is defined by Bronstein et al. [2021] as

$$\mathbf{h}_u = \phi\left(\mathbf{x}_u, \bigoplus_{v \in \mathcal{N}_u} \psi(\mathbf{x}_u, \mathbf{x}_v)\right)$$

for $\psi$ the *message function*, $\phi$ the *readout function* and $\oplus$ a permutation-invariant *aggregation function*. This GNN 'template' can be instantiated in many ways, with different choices of $\phi$, $\psi$ and $\oplus$ yielding popular architectures such as GCNs [Kipf and Welling, 2017] and GATs [Veličković et al., 2018].

### 2.1    To learn a complex aggregator is to learn a commutative monoid homomorphism

So we've seen that, in order to define a GNN, we must define a permutation-invariant aggregator $\oplus$ over its messages. But how can we characterise a permutation-invariant aggregator in general?

In abstract algebra (and in functional programming), a permutation-invariant aggregator over a set can be described as (maps into and out of) a ***commutative monoid***. A commutative monoid $(M, \oplus, e_\oplus)$ is a set $M$ equipped with a commutative, associative binary operator $\oplus : M \times M \to M$ and an identity element $e_\oplus \in M$ – in other words, an instance of the following Haskell typeclass, satisfying the identities to the right for all `x y z :: a` (see Snippet 1 in Appendix I for a Python version):

```
class CommutativeMonoid a =          x <> e == e
  e :: a                             x <> y == y <> x
  <> :: a -> a -> a                  x <> (y <> z) == (x <> y) <> z
```

Intuitively, commutative monoids over a set $M$ are 'operations you can use to reduce a multiset, whose members are in $M$, to a single value'. These include GNN aggregators, like *sum-aggregation* $(\mathbb{R}^n, +, \mathbf{0})$ and *max-aggregation* $(\mathbb{R}^n, \max, \mathbf{0})$. Indeed, Dudzik and Veličković [2022] observe that, for the aggregation function $\oplus$ of a GNN to be well-behaved (in the sense of respecting the axioms of the multiset monad), it must form a commutative monoid $(S, \oplus, e_\oplus)$ over some subspace $S$ of $\mathbb{R}^n$.

The vast majority of GNNs choose a fixed permutation-invariant function $\oplus$ (or fixed combinations of them [Corso et al., 2020]). While some research [Pellegrini et al., 2020, Li et al., 2020] has explored

aggregation functions with *learnable parameters*, these functions are only very weakly parameterised, and give us limited additional expressivity.

For problems where we can anticipate the kind of aggregation function we might need, this approach works well: indeed, choosing a commutative monoid that *aligns* with the algorithm we want our GNN to learn can improve performance both in and out of distribution [Veličković et al., 2019]. But there are many problems (e.g. those involving learning aggregations over representations of discrete values, or representations encoding many different types of data) for which these monoids may not always be the most natural choice for the aggregation we're trying to learn. So in such cases, $\psi$ and $\phi$ must take on some of the work of mapping our representations into and out of a space where $\oplus$-aggregation makes sense.

Formally, suppose we use a GNN equipped with a fixed commutative monoid aggregator $(F, \oplus, e_\oplus)$, on a problem for which the 'true' aggregation we want to perform is the commutative monoid $(M, *, e_*)$ over the GNN's latent space. What would it take for our GNN to perform $M$-aggregation?

**Proposition 1.** *Let $(M, *, e_*)$ and $(F, \oplus, e_\oplus)$ be commutative monoids. Then for functions $g : M \to F$ and $h : F \to M$,*

$$\mathop{\Large *}_{x \in X} x = h\left(\bigoplus_{x \in X} g(x)\right)$$

*for all finite multisets $X$ of $M$, if and only if $h$ is both a left inverse of $g$ and a surjective monoid homomorphism from $\langle g(M) \rangle \subseteq F$[1] to $M$.*

Now, given Proposition 1 above (proven in Appendix A), suppose we had a trained GNN, parameterised by $\phi : \mathbb{R}^k \times F \to \mathbb{R}^k$ and $\psi : \mathbb{R}^k \times \mathbb{R}^k \to F$, with a fixed $F$-aggregator. Suppose this GNN has *learned to imitate the $M$-aggregation commutative monoid*. We will model this property as there existing functions $\phi' : \mathbb{R}^k \times M \to \mathbb{R}^k$, $\psi' : \mathbb{R}^k \times \mathbb{R}^k \to M$, $g : M \to F$ and $h : F \to M$ such that

- $\phi(\mathbf{x}_u, \mathbf{m}_{\mathcal{N}(u)}) = \phi'(\mathbf{x}_u, h(\mathbf{m}_{\mathcal{N}(u)}))$
- $\psi(\mathbf{x}_u, \mathbf{x}_v) = g(\psi'(\mathbf{x}_u, \mathbf{x}_v))$

and $*_{x \in X}\, x = h(\bigoplus_{x \in X} g(x))$ for all finite multisets $X$ of $M$.

(Observe that this implies the following:

$$\phi\left(\mathbf{x}_u, \bigoplus_{v \in \mathcal{N}_u} \psi(\mathbf{x}_u, \mathbf{x}_v)\right) = \phi'\left(\mathbf{x}_u, h\left(\bigoplus_{v \in \mathcal{N}_u} g(\psi'(\mathbf{x}_u, \mathbf{x}_v))\right)\right)$$

$$= \phi'\left(\mathbf{x}_u, \mathop{\Large *}_{v \in \mathcal{N}_u} \psi'(\mathbf{x}_u, \mathbf{x}_v)\right)$$

for all nodes $u, v$ in graphs $G$.)

Hence $h$ is a surjective monoid homomorphism from $\langle g(M) \rangle$ to $M$ (i.e. $M$ is a subquotient of $F$).

So at a high level, *for a GNN with aggregator $F$ to imitate an aggregator $M$, it must learn a function that can decompose into a surjective monoid homomorphism from a submonoid of $F$ to $M$.*

## 2.2 Limitations on expressivity and generalisation for constructed aggregators

Given this result, what are the implications for prior and present work?

As has been seen in [Veličković et al., 2019, Sanchez-Gonzalez et al., 2020], it's clear that if our fixed commutative monoid $F$ is aligned with a target monoid $M$ for the problem we want to solve – intuitively, 'if the homomorphism doesn't have to do much work' – then we can easily learn to imitate $M$. Indeed, if the target homomorphism is linear, and we have appropriate training set coverage, then by [Xu et al., 2020] it may well generalise out-of-distribution – a result that holds (to an extent) in the case of learning to imitate path-finding algorithms such as Bellman-Ford [Veličković et al., 2019].

But there are many cases where $M$ is more complex, and there is no commonly-used fixed aggregator $F$ for which we can simply apply a linear homomorphism to get from $F$ to $M$. One such example is

---

[1] $\langle g(M) \rangle$ denotes the submonoid of $F$ generated by $g(M)$.

the problem of *finding the $2^{nd}$-minimum element in a set*. Here, the desired monoid $M$ is as follows: (Snippet 2)

```
type M = (Int, Int)                  secondMinimum :: [Int] -> Int
instance CommutativeMonoid M where   secondMinimum = dec . agg . map enc
  e = (infinity, infinity)             where
  (a1, a2) <> (b1, b2) = (c1, c2)        enc x = (x, infinity)
    where c1:c2:_ =                       agg = reduce (<>)
      sort [a1, a2, b1, b2]               dec (_, x2) = x2
```

Observe that, for this monoid, there is no such $F$ (e.g. sum, max, min, mean) for which there is a trivial choice of homomorphism from $F$ to $M$.

In principle, there exists an $F$ from which it is possible to construct a homomorphism to $M$: by [Zaheer et al., 2017] and [Xu et al., 2019a], for any $(M, *, e_*)$ with $M \subseteq \mathbb{Q}^n$, there exists a surjective monoid homomorphism $h$ from $(\mathbb{R}^n, +, 0)$ to $(M, *, e_*)$. But Wagstaff et al. [2019] show that this guarantee may require an $h$ that is highly discontinuous, and therefore not only hard to learn in-distribution[2], but fully misaligned with the assumptions of the universal approximation theorem. Further, as $\text{dom}(h) = \langle g(M) \rangle$, we are not learning a function whose domain is a bounded set, so we have little hope of generalising out-of-distribution. Indeed, we demonstrate in Section 3.1 that all common fixed aggregators fail to learn the $2^{nd}$-minimum problem, both in and out of distribution.

Similarly, Cohen-Karlik et al. [2020] show that sum-aggregators as implemented in [Zaheer et al., 2017] (i.e. maps into and out of the $(\mathbb{R}^n, +, 0)$ commutative monoid) require $\Omega(\log 2^n)$ neurons to learn the parity function over sets of size $n$. Intuitively, the crux of their proof is that *the homomorphism the aggregator would have to learn from $(\mathbb{R}^n, +, 0)$ to the parity monoid is a periodic function with unbounded domain*. Similar arguments hold for all aggregation tasks involving modular counting.

## 2.3 Fully learnable recurrent aggregators and their limitations

We will now take a step back from homomorphisms, and try to discover a more flexible aggregator. An emerging narrative within deep learning is that of *representations as types* [Olah, 2015]. If we view the construction of neural networks as the construction of differentiable, parameterised pure functional programs, many of the design patterns commonly used in deep learning correspond to higher-order functions commonly used in functional programming (FP). This paradigm has proven valuable in recent times, embodied by deep learning frameworks such as JAX [Bradbury et al., 2018].

In FP, a simple way to aggregate a multiset of elements is to represent them as a list and *fold* over it:[3] (Snippet 3)

```
fold :: (a -> b -> b) -> b -> [a] -> b
fold f z [] = z
fold f z (x:xs) = f x (fold f z xs)
```

And in some sense, a recurrent neural network (RNN) is simply a fold over a list, parameterised by a learnable accumulator `f` and a learnable initialisation element `z`:[4] (Snippet 4)

```
rnnCell :: Learnable             rnn :: Learnable
  (Vec R h1 -> Vec R h2 -> Vec R h2)   ([Vec R h1] -> Vec R h2)
initialState :: Learnable (Vec R h2)   rnn = fold rnnCell initialState
```

Hence a natural way to construct a *learnable* aggregator over multisets could be to use an RNN – a 'learnable fold' – and to somehow ensure it is permutation-invariant.

Indeed, this approach has been used for permutation-invariant set aggregation, with Murphy et al. [2019] enforcing permutation-invariance by design by taking the average of an RNN applied

---

[2]Suppose $f : X \to Y$ is a model trained to learn $h : X \to Y$ given a training set $\{(x_i, y_i)\}_{i=1}^n \subseteq D$ for $y_i = h(x_i)$ and $D$ the support of the training distribution. Now, for some loss function $L : Y \times Y \to \mathbb{R}$, we say that $f$ has *learned $h$ in-distribution* if $\mathbb{E}_{\mathbf{x} \sim \mathcal{D}}[L(f(x), h(x))]$ is small, and that $f$ has *learned $h$ out-of-distribution* if if $\mathbb{E}_{\mathbf{x} \sim \mathcal{P}}[L(f(x), h(x))]$ is small for distributions $\mathcal{P}$ over $X \backslash D$.

[3]Note that $a \to b \to b$ is an equivalent way (via **currying**) of specifying a function $a \times b \to b$.

[4]Note that an RNN can also be viewed as a map to the carrier set of the monoid of **endofunctions** (i.e. functions from a set to itself – in this case, from `b` to `b`) *under composition*: see Appendix B for details.

to all permutations of its input, and Cohen-Karlik et al. [2020] regularising RNNs $f$ towards permutation-invariance by adding a pairwise regularisation term $L_{swap}(\mathbf{x}_1, \mathbf{x}_2) = (f(f(\mathbf{s}, \mathbf{x}_1), \mathbf{x}_2) - f(f(\mathbf{s}, \mathbf{x}_2), \mathbf{x}_1))^2$ (which we motivate through the lens of commutative monoids in Appendix B).

Recurrent aggregators have also occasionally seen use in GNNs [Hamilton et al., 2017, Xu et al., 2018], but they are scarcely used despite their competitive performance. We assume RNNs likely remain unpopular as a GNN aggregator due to their *depth*. Indeed, observe that an $N$-layer GNN equipped with a recurrent aggregator has (worst-case) depth $O(VN)$. By contrast, the same GNN equipped with a fixed aggregator has (worst-case) depth $O(N)$. And as many graphs on which we want to deploy GNNs can have upwards of 100,000 nodes [Hu et al., 2020], the same problems of *efficiency* and *maximum dependency length* observed by Vaswani et al. [2017] when using RNNs for sequence transduction also hold when using RNNs for graph message aggregation.

### 2.4 A compromise: fully learnable commutative monoids

So, if recurrent aggregators are too deep, is there any way to get a fully learnable aggregator? We've considered the *fixed-aggregator* approach, where we learn maps into and out of the carrier set of a pre-determined commutative monoid. We've considered the *recurrent-aggregator* approach, where we represent multisets as lists and implement aggregation as a *learnable fold over lists*.[5] But another way to represent multisets in FP is as a *balanced binary tree*, over which aggregation is implemented as a fold parameterised by a commutative monoid. So what if we implemented aggregation as a *learnable fold over a balanced binary tree*? Or in other words, what if, instead of learning maps into and out of some commutative monoid, we simply *learn the commutative monoid itself*?

Let's make precise what exactly we mean by 'learning a commutative monoid' for use in a GNN. Recall that a commutative monoid $(M, \oplus, e_{\oplus})$ is defined by its carrier set $M$, its binary operation $\oplus$ and its identity element $e_{\oplus}$. So given some learnable *commutative, associative* binary operator $\oplus$ (written `binOp :: Learnable (Vec R h -> Vec R h -> Vec R h)`), and some learnable identity element $e_{\oplus}$ (written `identity :: Learnable (Vec R h)`), we can define a ***learnable commutative monoid*** over some learned embedding space (in other words, a subset of $\mathbb{R}^h$): (Snippet 5)

```
type HiddenState = Vec R h
instance CommutativeMonoid HiddenState where
  e = identity; <> = binOp
```

Thus, our aggregation function can be specified simply, as $\bigoplus_x x$, or

```
aggregate :: Learnable ([HiddenState] -> HiddenState)
aggregate = reduce (<>)
```

Note that, here, the carrier set is implicit – when used in a GNN, we expect the message function (i.e. the producer of the elements to be aggregated) to learn a 'return type' representation whose members are elements of this implicit carrier set, and similarly for the 'input type' of the readout function.

Now, why do we care about this at all? Indeed, if we implement `reduce` as a `fold`, we're no better off than if we just used a recurrent aggregator. But consider the computation graph (or rather, computation *binary tree*) of such an aggregation $x_1 \oplus (x_2 \oplus (x_3 \oplus x_4))$. By Tamari's theorem [Tamari, 1962], the associativity of $\oplus$ means that the result of evaluating this computation tree is *invariant under rotations of nodes in the tree*. Therefore, in order to minimise the depth of the computation, we can rewrite our reduction as a balanced binary tree: $(x_1 \oplus x_2) \oplus (x_3 \oplus x_4)$ (see Appendix D). And by doing so, for $V$ elements to aggregate, we obtain a network with $O(V)$ applications of $\oplus$ and $O(\log V)$ depth – an exponential improvement over our $O(V)$-depth recurrent aggregators.

### 2.5 Commutative, associative binary operators for learnable commutative monoids

So, *given a commutative, associative binary operator*, we can get our learnable commutative monoid with $O(\log V)$ depth. But how do we construct such an operator in the first place? As with permutation-invariant RNNs, we have two options: either we construct an operator that *strongly enforces* the axioms of commutativity and associativity by construction, or we construct some arbitrary binary operator and *weakly enforce* the axioms through regularisation.

---

[5] Alternatively, we can see this, as in Appendix B, as learning maps into and out of the carrier set of the monoid of endofunctions.

**Strong enforcement.** While some research has been conducted into learning algebraic structures with *strongly enforced* axioms [Abe et al., 2021, Martires, 2021], these approaches reduce to *learning maps into and out of a fixed aggregator*.[6] We observe that, while we can strongly enforce commutativity in any binary operator $f(x, y)$ by symmetrising it to $g(x, y) = \frac{f(x,y) + f(y,x)}{2}$, we found no such construction for associativity which doesn't sacrifice expressivity.

So given this, and given the importance of *gating* [Tallec and Ollivier, 2018] in neural networks applied over long time horizons, we can construct a simple *strongly commutative* binary aggregator (**Binary-GRU**) by symmetrising a GRU [Cho et al., 2014]: (Snippet 6)

```
binaryGRU :: Learnable (Vec R h -> Vec R h -> Vec R h)
binaryGRU v1 v2 = do
  g <- new gruCell (InputDim h) (HiddenDim h)
  return (g v1 v2 + g v2 v1) / 2
```

**Weak enforcement.** Alternatively, just as we saw with recurrent aggregators in Section 2.3, for a learnable binary operator $\oplus : \mathbb{R}^n \to \mathbb{R}^n \to \mathbb{R}^n$ we could *weakly enforce* commutativity and associativity through regularisation losses $L_{comm}(\mathbf{x}, \mathbf{y}) = \lambda_{comm}|(\mathbf{x} \oplus \mathbf{y}) - (\mathbf{y} \oplus \mathbf{x})|^2$ and $L_{assoc}(\mathbf{x}, \mathbf{y}, \mathbf{z}) = \lambda_{assoc}|(\mathbf{x} \oplus (\mathbf{y} \oplus \mathbf{z})) - ((\mathbf{x} \oplus \mathbf{y}) \oplus \mathbf{z})|^2$ (for implementation details, see Appendix E).

Now, by applying $L_{assoc}$ to Binary-GRU, we obtain a *strongly commutative*, *weakly associative* binary operator (**Binary-GRU-Assoc**).[7]

## 3 Assessing the utility of learnable commutative monoids

Now, we've seen three types of aggregator: fixed aggregators, recurrent aggregators and learnable commutative monoids. In order to explore their trade-offs in terms of *expressivity*, *generalisation* and *efficiency*, we conduct a range of experiments comparing the performance of

- *state-of-the-art fixed aggregators* (such as sum-aggregation [Zaheer et al., 2017], max-aggregation [Veličković et al., 2019] and PNA [Corso et al., 2020]),
- *recurrent aggregators* (specifically GRUs [Cho et al., 2014]), and
- *learnable commutative monoid (LCM) aggregators* (using the Binary-GRU and Binary-GRU-Assoc learnable operators as described in Section 2.5)

on the following synthetic and real-world problems:

**2nd-minimum.** We test fixed aggregators, recurrent aggregators and learnable commutative monoids on the problem of finding the *second-smallest* element in a set of binary-encoded integers. As observed in Section 2.2, this task is a synthetic aggregation problem with an 'unusual' commutative monoid, in that it doesn't align well with common fixed aggregators. Therefore, we expect this task to be a standard problem for which learnable aggregators would outperform any commonly-used fixed aggregator, especially out-of-distribution.

**PNA synthetic benchmark.** We then proceed to test the in-distribution performance of our aggregators on the synthetic dataset presented in [Corso et al., 2020]. This dataset consists of aggregator-heavy, classical graph problems that are mostly aligned with the aggregators used to construct PNA. Thus, we expect PNA (and the relevant fixed aggregators) to perform strongly here, potentially even out-of-distribution. But while our learnable aggregators don't necessarily have the inductive bias to approximate these monoids well over an *unbounded* domain, we expect them to perform competitively at learning the relevant monoids in-distribution.

**PNA real-world benchmark.** Finally, we test our aggregators on the real-world dataset presented in [Corso et al., 2020], consisting of chemical (ZINC and MolHIV) and computer vision (CIFAR10 and MNIST) datasets from the GNN benchmarks of Dwivedi et al. [2020] and Hu et al. [2020]. In contrast to the algorithmic tasks in the synthetic benchmark, we expect these real-world problems to contain 'unusual' target monoids: for both molecular and computer vision problems, it is likely that

---

[6]i.e. choosing an algebraic structure (e.g. the Abelian group $(\mathbb{R}^n, +, \mathbf{0})$) and learning maps between the model's latent space and that structure.

[7]Note that we can instantiate this operator with different values of the *regularisation parameter* $\lambda_{assoc}$ (hereafter referred to as $\lambda$) by which we scale the associativity loss.

our GNN will learn complex representations whose most natural monoid is not the image of a simple homomorphism from any common fixed aggregator. Therefore, we expect fully learnable aggregators (GRU and LCMs) to outperform fixed aggregators on this benchmark.

Training details for all experiments are provided in Appendix F. Notably, for all uses of learnable aggregators, we randomly shuffle each batch of sequences before feeding it to the aggregator as a form of *regularisation through data augmentation*.

## 3.1 2nd-minimum

For this experiment, we compared fixed (sum, max, PNA), recurrent (GRU) and LCM (Binary-GRU) aggregators on the synthetic **2nd-minimum** set aggregation problem. In order to evaluate the effects of *regularisation towards algebraic axioms* on the performance of LCM aggregators, we also tested Binary-GRU-Assoc, sweeping over values of the regularisation parameter $\lambda$ from $10^0$ to $10^{-7}$.

### 3.1.1 Experimental details

For *training data*, we used 65,536 multisets of integers $\sim U(0, 255)$ of size $\sim U(1, 16)$. For *validation data*, we used 1,024 multisets of integers $\sim U(0, 255)$ of size 32. For *evaluation data*, we used 1,024 multisets of integers $\sim U(0, 255)$ of size $l$, for $l \in [1, 200]$. We used a standard multiset-aggregation architecture $f(\mathbf{X}) := \sigma(\psi(\bigoplus_{\mathbf{x} \in \mathbf{X}} \phi(\mathbf{x})))$ for $\oplus$ the aggregator being tested, and $\phi$ and $\psi$ MLPs. $f$ takes as input a vector of 8-bit binary-encoded integers (as in [Yan et al., 2020]), and returns a binary-encoded integer in $[0, 1]^8$. The full architecture (with details on integer embedding) is outlined in Appendix C.

### 3.1.2 Results and discussion

**Summary.** Recall that this problem was chosen for its comparatively unusual commutative monoid, which we do not expect aligns well with fixed aggregators. Indeed, we confirm this hypothesis: we see in Figure 1 that *fixed aggregators* fail to learn 2nd-minimum in-distribution, that *recurrent aggregators* learn 2nd-minimum near-perfectly in-distribution, generalising well out-of-distribution, and that *LCM aggregators* learn 2nd-minimum near-perfectly in-distribution and are competitive with recurrent aggregators out-of-distribution, while achieving an exponential speedup over recurrent aggregators on large sets. Furthermore, we observe that *regularising towards algebraic axioms* improves the performance of LCM aggregators both in and out of distribution.

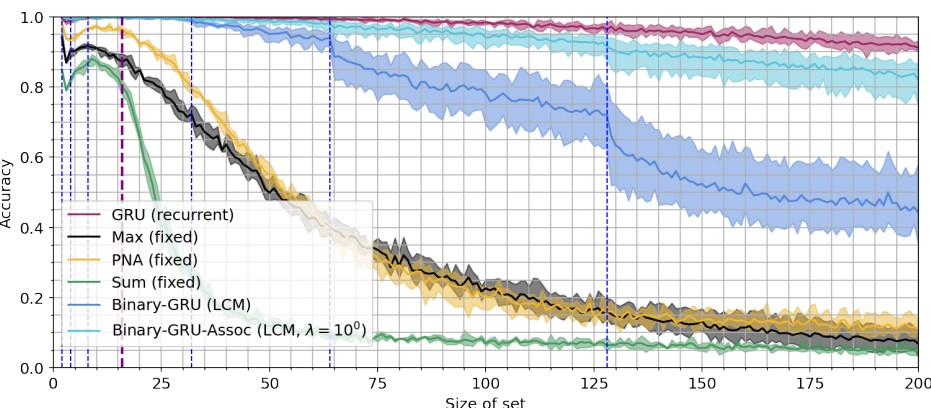

**Figure 1:** Generalisation performance for fixed (max, sum, PNA), recurrent (GRU) and LCM (Binary-GRU) aggregators, along with the best-performing regularised LCM aggregator (Binary-GRU-Assoc with $\lambda = 10^0$). The shaded region is bounded above and below by the maximum and minimum values across all runs. The vertical purple line denotes the maximum set size present in training data (16); the vertical blue lines denote powers of 2 (from $2^1$ to $2^7$). For detailed results, see Appendix G.

**In-distribution performance.** Examining Figure 1, observe that only the fully-learnable aggregators – GRU, Binary-GRU and Binary-GRU-Assoc – managed to learn $2^{nd}$-minimum near-perfectly in-distribution, with the next best performing aggregator being PNA.[8]

**Out-of-distribution performance (without regularisation).** Observe that, out-of-distribution, all learnable aggregators generalise near-perfectly up to size 32 (twice the size of the input). Beyond this point, while the performance of the recurrent aggregator decays slowly (reaching $0.912 \pm 0.017$ at size 200), the performance of the LCM quickly drops (reaching $0.287 \pm 0.068$ at size 200). Despite this, both learnable aggregators consistently outperform the fixed aggregators out-of-distribution. Furthermore, out of the fixed aggregators, we see that the sum-aggregator's performance plateaus extremely quickly, a result we may attribute to *the domain of the learned homomorphism from the sum-aggregator being an unbounded set* (see Section 2.2).

**Efficiency.** As hypothesised in Section 2.4, we see (in Appendix G, Figure 3) that LCMs are indeed exponentially faster than RNNs for large sets: for $n = 20$, Binary-GRU-Assoc takes $48.2 \pm 0.4$ seconds per epoch, and GRU takes $46.6 \pm 0.5$ seconds per epoch, while for $n = 200$, Binary-GRU-Assoc takes $79.4 \pm 0.5$ seconds per epoch, and GRU takes $397.2 \pm 1.3$ seconds per epoch.

**Regularisation towards associativity.** We show the results from the best-performing regularised LCM aggregator ($\lambda = 10^0$) in Figure 1 and Table 2. Although the unregularised Binary-GRU performs better than all fixed aggregators, observe that the regularised Binary-GRU-Assoc outperforms its unregularised sibling both in and out of distribution, and achieves generalisation performance competitive with GRU. Furthermore, observe that the sudden performance drops experienced by Binary-GRU when the size of the set reaches a power of two (i.e. when the depth of the aggregation tree increases) are noticeably dampened for Binary-GRU-Assoc, suggesting that regularisation towards associativity helps prevent overfitting to a particular maximum aggregation tree height. For interest, we present the full results of the regularisation parameter sweep in Figure 4 in Appendix G.

## 3.2 PNA synthetic benchmark

For this experiment, we trained recurrent (GRU) and LCM (Binary-GRU, Binary-GRU-Assoc) aggregators on the synthetic benchmark from [Corso et al., 2020], comparing against the fixed-aggregator baselines presented there (for GATs [Veličković et al., 2018], GCNs [Kipf and Welling, 2017], GINs [Xu et al., 2019b] and MPNNs [Gilmer et al., 2017] with sum and max aggregators).

### 3.2.1 Experimental details

In the PNA paper [Corso et al., 2020], experiments testing fixed aggregators (sum, max, PNA) are conducted on a custom GNN architecture centred around an MPNN layer with dimension 16, split into four towers each with hidden dimension 4. As we hypothesise that the low dimensionality of these towers could harm the expressivity of learnable aggregators, we test our learnable aggregators both in MPNNs of hidden dimension 16, with four towers of hidden dimension 16, and in MPNNs of hidden dimension 128, with one tower of hidden dimension 128.

### 3.2.2 Results and discussion

**Summary.** Recall that this dataset consists of aggregator-heavy classical graph problems[9] that are mostly aligned with the aggregators used to construct PNA. So, as expected, we see in Table 1 that PNA outperforms all other aggregators tested on the dataset in-distribution. But observe that, on these problems, *our asymptotically more efficient LCMs are competitive with and sometimes beat GRUs* – and indeed, on the node-based problems in the dataset, our LCMs are as strong as PNA.

In Appendix G, we observe the surprising result that LCMs are more stable than PNA out-of-distribution (OOD), and that regularising LCMs towards associativity improves OOD performance at the cost of impairing performance in-distribution. We also discuss the *effects of increasing dimensionality* on fixed aggregator performance, through the lens of commutative monoid homomorphisms.

---

[8]Note that, out of the fixed aggregators, PNA was the only one to achieve near-perfect accuracy on the *training* dataset, with a maximum training accuracy of around 0.997.

[9]three node-based algorithmic tasks (single-source shortest paths, eccentricity and computing the Laplacian of node feature vectors) and three graph-based algorithmic tasks (connectedness, diameter and spectral radius)

**In-distribution performance.** Observe in Table 1 that, while PNA beats all other aggregators tested, our learnable aggregators perform competitively in-distribution, with all learnable aggregators (GRU, Binary-GRU and Binary-GRU-Assoc) beating all single-aggregator (i.e. non-PNA) architectures. Interestingly, our Binary-GRUs perform better than the corresponding GRUs: perhaps their *inductive bias towards commutativity* helps us learn in-distribution.

| Model | Avg score | Node tasks | | | Graph tasks | | |
|---|---|---|---|---|---|---|---|
| | | SSSP | Ecc | Lap feat | Conn | Diam | Spec rad |
| GCN | -2.05 | -2.16 | -1.89 | -1.60 | -1.69 | -2.14 | -2.79 |
| GAT | -2.26 | -2.34 | -2.09 | -1.60 | -2.44 | -2.40 | -2.70 |
| GIN | -1.99 | -2.00 | -1.90 | -1.60 | -1.61 | -2.17 | -2.66 |
| MPNN (sum) | -2.50 | -2.33 | -2.26 | -2.37 | -1.82 | -2.69 | -3.52 |
| MPNN (max) | -2.53 | -2.36 | -2.16 | -2.59 | -2.54 | -2.67 | -2.87 |
| PNA-16 | -3.04 | **-2.99** | -2.81 | -2.83 | **-2.91** | -2.98 | **-3.71** |
| PNA-128 | **-3.09** | -2.94 | **-2.88** | -3.82 | -2.42 | **-3.00** | -3.48 |
| GRU | -2.91 | -2.84 | -2.71 | -3.73 | -2.20 | -2.88 | -3.11 |
| Binary-GRU | -3.00 | -2.85 | -2.77 | **-3.87** | -2.34 | -2.88 | -3.29 |
| Binary-GRU-Assoc | -2.95 | **-2.99** | **-2.88** | -2.92 | -2.62 | -2.92 | -3.37 |

**Table 1:** Mean $\log_{10}(MSE)$ on the PNA test dataset

**Per-task performance.** We present the per-task performance of all 128-dimensional aggregators (together with fixed-aggregator baselines) in Table 1. Observe that, in fact, Binary-GRU-Assoc outperforms Binary-GRU in all tasks apart from the the graph Laplacian.

Furthermore, while learnable aggregators do not perform as strongly as fixed aggregators on whole-graph tasks, they perform as well as or better than fixed aggregators for node-based tasks. This may be because the benchmark implementation for whole-graph tasks uses a *sum-aggregator* over the readout values: it is likely difficult to learn a homomorphism from the sum aggregator to the complex latent-space monoid learned by the LCM, and perhaps fixed aggregators provide an inductive bias towards learning representations for which it is easier to map to and from the sum-aggregation monoid.

## 3.3 PNA real-world benchmark

For this experiment, we trained recurrent (GRU) and LCM (Binary-GRU) aggregators on the real-world benchmark from Corso et al. [2020], containing two molecular graph property prediction datasets (ZINC and MolHIV) and two superpixel graph classification datasets (CIFAR10 and MNIST). Note that, due to limitations on compute resources, we were not able to perform a regularisation parameter sweep to test Binary-GRU-Assoc. The GNN architecture used here is identical to that in [Corso et al., 2020], except that, for learnable aggregators, all MPNN towers have the same dimensionality as the MPNN itself (i.e. we do not *divide* the towers).

### 3.3.1 Results and discussion

**Summary.** Recall that the real-world benchmark has complex problems that do not necessarily align with common fixed aggregators. We observe in Figure 2 that, while PNA in general outperforms all other aggregators on property prediction problems over small molecular graphs, the more expressive GRU substantially outperforms PNA for the (more discrete) task of image classification. Also, note that the (asymptotically efficient) Binary-GRU LCM provides a good trade-off between these two aggregators, being the *second-best aggregator* for all but two problems. Finally, we see that learnable aggregators appear particularly powerful on problems involving *graphs with edge features*.

**Molecular datasets.** Observe that PNA is the strongest aggregator over both the ZINC dataset without edge features and the HIV dataset – indeed, due to the continuous nature of the properties we want to estimate in these datasets, it seems likely that the 'natural' monoids for aggregation over graphs in these datasets would align well with fixed aggregators.

**Image datasets.** By contrast, we observe that GRU-aggregators are the strongest when testing on image data, likely as their expressivity lets them easily learn a complex, perhaps more discrete aggregation function. And while Binary-GRU does not do quite as well as GRU here, in all but one case it outperforms PNA on this problem.

**Edge features.** Finally, observe that, if we add edge features to ZINC, GRU outperforms PNA – and comparing results on the CIFAR-10 dataset with and without edge features, the average accuracy

| | | Zinc (MAE) | | | | CIFAR10 (Acc) | | | | MNIST (MAE) | | | | MolHIV (%ROC-AUC) | |
|---|---|---|---|---|---|---|---|---|---|---|---|---|---|---|---|
| | Model | No edge features | std | Edge features | std | No edge features | std | Edge features | std | No edge features | std | Edge features | std | No edge features | std |
| Dwivedi et al, Xu et al | MLP | 0.710 | 0.001 | | | 56.01 | 0.90 | | | 94.46 | 0.28 | | | | |
| | GCN | 0.469 | 0.002 | | | 54.46 | 0.10 | | | 89.99 | 0.15 | | | 76.06 | 0.97 |
| | GIN | 0.408 | 0.008 | | | 53.28 | 3.70 | | | 93.96 | 1.30 | | | 75.58 | 1.40 |
| | DiffPoll | 0.466 | 0.006 | | | 57.99 | 0.45 | | | 95.02 | 0.42 | | | | |
| | GAT | 0.463 | 0.002 | | | 65.48 | 0.33 | | | 95.62 | 0.13 | | | | |
| | Monet | 0.407 | 0.007 | | | 53.42 | 0.43 | | | 90.36 | 0.47 | | | | |
| | GatedGCN | 0.422 | 0.006 | 0.363 | 0.009 | 69.19 | 0.28 | 69.37 | 0.48 | 97.37 | 0.06 | 97.47 | 0.13 | | |
| Corso et al | MPNN (sum) | 0.381 | 0.005 | 0.288 | 0.002 | 65.39 | 0.47 | 65.61 | 0.30 | 96.72 | 0.17 | 96.90 | 0.15 | | |
| | MPNN (max) | 0.468 | 0.002 | 0.328 | 0.008 | 69.70 | 0.55 | 70.86 | 0.27 | 97.37 | 0.11 | 97.82 | 0.08 | | |
| | PNA (no scalers) | 0.413 | 0.006 | 0.247 | 0.036 | 70.46 | 0.44 | 70.47 | 0.72 | 97.41 | 0.16 | 97.94 | 0.12 | 78.76 | 1.04 |
| | PNA | **0.320** | 0.032 | 0.188 | 0.004 | 70.21 | 0.15 | 70.35 | 0.63 | 97.19 | 0.08 | 97.69 | 0.22 | **79.05** | 1.32 |
| Ours | Binary-GRU | 0.340 | 0.003 | 0.175 | 0.003 | 69.61 | 0.18 | 71.86 | 0.26 | 97.79 | 0.20 | 98.11 | 0.07 | 77.37 | 1.11 |
| | GRU | 0.342 | 0.004 | **0.171** | 0.006 | **72.03** | 1.06 | **74.44** | 0.52 | 98.15 | 0.04 | **98.41** | 0.10 | 76.04 | 1.01 |

**Figure 2:** Results of learnable aggregators on the PNA real-world dataset, in comparison with those analysed by Corso et al. [2020]. Best results in bold-face, second-best in underline.

improvement for fixed aggregators when adding edge features is 0.34%, whereas the equivalent improvement for learnable aggregators is 2.33%. Learnable aggregators may be particularly strong on tasks with *edge features*, as making full use of them tends to require the learning of a more complex aggregation function.

## 4 Conclusions

In this work we have conducted a thorough study of aggregation functions within graph neural networks (GNNs), demonstrating both theoretically and empirically that many tasks of practical interest rely on a nontrivial integration of neighbourhoods (i.e. a nontrivial *commutative monoid*). This motivates the use of fully-learnable aggregation functions, but prior proposals based on RNNs had several shortcomings in terms of efficiency. Accordingly, we propose learnable commutative monoid (LCM) aggregators, which trade off the flexibility of RNNs with efficiency of fixed aggregators, producing a simple, yet empirically powerful, GNN aggregator with only $O(\log V)$ depth.

**Implications for GNN practitioners.** Based on our results, we present some suggestions to those using GNNs in practice:

- When **choosing a fixed aggregator** $F$ for a GNN architecture, consider the type of aggregation your problem is likely to involve – if it can be framed as a commutative monoid $M$, is it likely that a homomorphism can be learned from $F$ to $M$?

- For **graph problems manipulating discrete data**, or problems for which the aggregation required *doesn't align* with existing fixed aggregators, learnable aggregators may improve performance (especially out-of-distribution).

- When **choosing a learnable aggregator**, for problems over small graphs, recurrent aggregators will likely perform well – but if they prove too slow, you may wish to try a learnable commutative monoid aggregator.

- And **if your learnable aggregator is overfitting**, perhaps try *regularising* it towards the relevant axioms (e.g. invariance under pairwise swaps for recurrent aggregators, commutativity and associativity for learnable commutative monoids).

## Author Contributions

The research idea of exploring learnable commutative monoid (LCM) aggregators was originated by Euan Ong and steered by Petar Veličković. The experimental pipeline was developed and experiments were conducted by Euan, with oversight, mentorship and management from Petar. The formal analysis of fixed aggregators and its framing in terms of functional programming were originated by Euan, with advice from Petar. Both authors contributed to writing the paper and responding to reviewer feedback.

## Acknowledgements

We would like to thank Pietro Liò and Malcolm Scott for generously providing access to compute resources at short notice, without which this work would not be possible. We also thank Andrew Dudzik and Karl Tuyls for reviewing the paper prior to submission, and all our anonymous reviewers for their careful feedback.

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

## A Proof of Proposition 1

**Proposition 1.** *Let* $(M, *, e_*)$ *and* $(F, \oplus, e_\oplus)$ *be commutative monoids. Then for functions* $g : M \to F$ *and* $h : F \to M$,

$$\underset{x \in X}{*} x = h\left(\bigoplus_{x \in X} g(x)\right)$$

*for all finite multisets* $X$ *of* $M$, *if and only if* $h$ *is both a left inverse of* $g$ *and a surjective monoid homomorphism from* $\langle g(M)\rangle \subseteq F$[10] *to* $M$.

*Proof.* We proceed by cases.

($\to$) Suppose $*_{x \in X}\, x = h(\bigoplus_{x \in X} g(x))$ for all finite multisets $X$ of $M$.

When $X = \{x\}$, have that $h(g(x)) = x$ trivially, so $h$ must be a left inverse of $g$ (and is therefore surjective).

Now for $x, y \in \langle g(M)\rangle$, we want to show that $h(x \oplus y) = h(x) * h(y)$ and that $h(e_\oplus) = e_*$.

To show the former, observe that $x = \bigoplus_{a \in A} g(a)$ and $y = \bigoplus_{b \in B} g(b)$ for some finite multisets $A, B$ of $M$.

Now have that

$$
\begin{aligned}
h(x \oplus y) &= h\left(\left(\bigoplus_{a \in A} g(a)\right) \oplus \left(\bigoplus_{b \in B} g(b)\right)\right) \\
&= h\left(\bigoplus_{x \in A \uplus B} g(x)\right) \\
&= \underset{x \in A \uplus B}{*} x \\
&= \left(\underset{a \in A}{*} a\right) * \left(\underset{b \in B}{*} b\right) \\
&= h\left(\bigoplus_{a \in A} g(a)\right) * h\left(\bigoplus_{b \in B} g(b)\right) \\
&= h(x) * h(y)
\end{aligned}
$$

as desired.

To show the latter, observe that $h(e_\oplus) * h(f) = h(e_\oplus \oplus f) = h(f)$ for all $f \in F$. As $h$ is surjective, we have that $h(F) = M$, so $h(e_\oplus) * m = m * h(e_\oplus) = m$ for all $m \in M$, and $h(e_\oplus) = e_*$.

---

[10] $\langle g(M)\rangle$ denotes the submonoid of $F$ generated by $g(M)$.

($\leftarrow$) Suppose $h$ is a left inverse of $g$ and a surjective monoid homomorphism from $\langle g(M) \rangle$ to $M$. Then

$$
\begin{aligned}
h\left( \bigoplus_{x \in X} g(x) \right) &= h\left( \bigoplus_{i=1}^{n} g(x_i) \right) \\
&= h\left( f(x_1) \oplus \bigoplus_{i=2}^{n} g(x_i) \right) \\
&= h(g(x_1)) * h\left( \bigoplus_{i=2}^{n} g(x_i) \right) \\
&= x_1 * h\left( \bigoplus_{i=2}^{n} g(x_i) \right) \\
&= \ldots \\
&= \mathop{\text{\Large ✳}}_{i=1}^{n} x_i \\
&= \mathop{\text{\Large ✳}}_{x \in X} x
\end{aligned}
$$

as desired.

$\square$

## B  Motivating the conditions for permutation-invariance in RNNs

An alternative way to motivate the regularisation loss of Cohen-Karlik et al. [2020], through the lens of monoids, is to frame the recurrent aggregator as a monoid, and identify the conditions required for this monoid to be commutative.

Keeping in mind that 'RNNs are just learnable folds', we notice that *endofunctions form a monoid under composition*:

```
instance Monoid (a -> a) where
  e = id
  <> = (.)
```

and observing that, for instance,

```
  fold f z [x1, x2, x3]
= f x1 (f x2 (f x3 z))
= (f x1 . f x2 . f x3) z
= ($ z) (f x1 . f x2 . f x3)
= ($ z) (reduce (.) (map f [x1; x2; x3]))
```

we can rewrite `fold` as an aggregation over the composition monoid:

```
fold :: (a -> b -> b) -> b -> [a] -> b
fold f z = dec . reduce (.) . map enc
  where
    enc x = f x
    dec f = f z
```

Now, applying this to our recurrent aggregator, we have

```
rnn :: Learnable ([Vec R h1] -> Vec R h2)
rnn = dec . reduce (.) . map enc
  where
    enc x = rnnCell x
    dec f = f initialState
```

Observe that, for `rnn`, the carrier set of the composition (sub)monoid consists of functions `rnnCell x` for inputs `x` to the aggregation function. So, in order to enforce that this monoid is commutative, we must simply ensure that

```
    f <> g = g <> f
=> (rnnCell x1) . (rnnCell x2) = (rnnCell x2) . (rnnCell x1)
=> rnnCell x1 (rnnCell x2 h) = rnnCell x2 (rnnCell x1 h)
```

for all inputs `x1`, `x2` and all hidden states `h`.

## C  Architecture used for 2nd-minimum benchmark

We present Haskell pseudocode for the architecture used in the 2nd-minimum benchmark below.

```
h = 128

ofMlp :: Learnable (Vec R h -> Vec R h)
ofMlp = do
  dense <- new ofLinearLayer (In h) (Out h)
  return gelu . dense

intEmbedding :: Learnable (Vec Bool 8 -> Vec R h)
intEmbedding = toLearnable $ \int -> do
  one_vecs <- newList (Length 8) (Of (learnableParameter (Dim h)))
  zero_vecs <- newList (Length 8) (Of (learnableParameter (Dim h)))
  return
    [ one*i + zero*(1-i)
    | (i, one, zero) <- zip3 int oneVecs zeroVecs]

enc :: Learnable (Vec Bool 8 -> Vec R h)
enc = do
  mlp <- new ofMlp
  return mlp . intEmbedding

agg :: Learnable ([Vec R h] -> Vec R h)
-- Implementation-dependent

dec :: Learnable (Vec R h -> Vec R 8)
dec = do
  mlp <- new ofMlp
  dense <- new ofLinearLayer (In h) (Out h)
  return sigmoid . dense . mlp

net :: Learnable ([Vec Bool 8] -> Vec R 8)
net = dec . agg . map enc
```

## D  Implementing binary tree aggregation for learnable commutative monoids

More precisely, given a learnable commutative monoid operator `<>` and a function `toBalancedTree` which takes a list of elements and returns a balanced `Tree` whose leaves contain these elements, we aggregate in the following way:

```
data Tree a = Lf a | Nd Tree Tree
toBalancedTree :: [a] -> Tree a

fold :: (a -> a -> a) -> Tree a -> a
fold f = \case
  Nd l r -> f (fold f l) (fold f r)
  Lf m -> m
```

```
aggregate :: Learnable ([LearnableMonoid] -> LearnableMonoid)
aggregate = fold (<>) . toBalancedTree
```

## E   Implementing regularisation losses for learnable commutative monoids

Observe that, for any learnable binary operator

```
(<>) :: Learnable (Vec R h -> Vec R h -> Vec R h)
```

aggregating over a tree of messages (of type `Tree (Vec R h)`), we can construct regularisation losses that penalise the operator for violating commutativity and associativity each time it is applied:

```
-- Computes getLossesAtNode at every node in the tree,
-- returning a list of the results.
accumLosses :: ((Tree (Vec R h)) -> [R]) -> (Tree (Vec R h)) -> [R]
accumLosses getLossesAtNode = \case
  Nd a b ->
    getLossesAtNode (Nd a b) :
      (accumLosses getLossesAtNode a ++ accumLosses getLossesAtNode b)
  Lf -> _

commLoss :: (Tree (Vec R h)) -> R
commLoss = mean . accumLosses getLossesAtNode
  where getLossesAtNode = \case
    Nd a b -> [|(a <> b) - (b <> a)|**2]
    Lf -> []

assocLoss :: (Tree (Vec R h)) -> R
assocLoss = mean . accumLosses getLossesAtNode
  where
    loss a b c = |((a <> b) <> c) - (a <> (b <> c))|**2
    getLossesAtNode = \case
      Nd (Nd a b) (Nd c d) ->
        [loss (aggregate a) (aggregate b) (aggregate c),
         loss (aggregate b) (aggregate c) (aggregate d)]
      Nd (Nd a b) (Lf c)   ->
        [loss (aggregate a) (aggregate b) c]
      _ -> []

aggregateWithLoss :: Learnable ([LearnableMonoid] -> LearnableMonoid)
aggregateWithLoss xs = aggregate tree
  with extraLosses = [commLoss tree, assocLoss tree]
  where tree = toBalancedTree xs
```

## F   Training details for experiments

On every experiment, for each model, we performed 3 training runs with different seeds; for each run we used a validation set to choose the highest-performing checkpoint for evaluation.

**2nd-minimum.**   We trained each aggregator with the Adam optimiser for 1,000 epochs, with batch size 32 and learning rate $1e - 4$.

**PNA synthetic benchmark.**   We trained each aggregator for 1,000 epochs. To ensure convergence, 16-dimensional models were trained with a learning rate of $10^{-3}$ as in Corso et al. [2020], and 128-dimensional models were trained with a learning rate of $10^{-4}$. All other hyperparameters were as in Corso et al. [2020].

**PNA real-world benchmark.**   All hyperparameters (including training time) are as in [Corso et al., 2020].

# G   Detailed results for the 2nd-minimum benchmark

We present more detailed results for the 2nd-minimum benchmark below:

- Table 2 contains in-distribution and out-of-distribution results for all aggregators tested.
- Figure 3 presents network efficiency against set size for all aggregators tested.
- Figure 4 presents the full results of the regularisation parameter sweep for Binary-GRU-Assoc.

As a side note, when training the non-regularised Binary-GRU aggregators, we observed that while associativity regularisation loss increased initially, it started *decreasing* as the GNN's training accuracy began to plateau. This potentially hints at the model's learning trajectory: one might hypothesise that the point at which the loss decreases is the point at which the model shifts from memorisation to learning a parsimonious algorithm that generalises.

| Type | Aggregator | ID accuracy | OOD accuracy | |
|---|---|---|---|---|
| | | $n \in [1, 16]$ | $n = 32$ | $n = 200$ |
| Recurrent | GRU | **0.996 ± 0.001** | **0.998 ± 0.001** | **0.912 ± 0.017** |
| LCM | Binary-GRU-Assoc | **0.997 ± 0.002** | **0.997 ± 0.002** | 0.822 ± 0.064 |
| LCM | Binary-GRU | **0.997 ± 0.001** | **0.992 ± 0.005** | 0.443 ± 0.122 |
| Fixed | PNA | 0.961 ± 0.003 | 0.794 ± 0.012 | 0.110 ± 0.027 |
| Fixed | Max | 0.901 ± 0.007 | 0.723 ± 0.025 | 0.069 ± 0.039 |
| Fixed | Sum | 0.845 ± 0.010 | 0.261 ± 0.020 | 0.045 ± 0.011 |

**Table 2:** Accuracy (the fraction of multisets at each size for which the 2nd-minimum is correctly identified) for fixed, recurrent and LCM aggregators, along with the best-performing regularised LCM aggregator (Binary-GRU-Assoc with $\lambda = 10^0$).

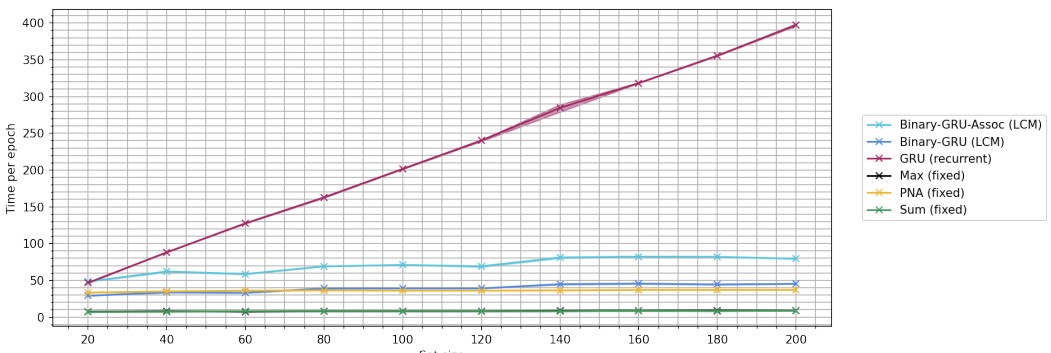

**Figure 3:** Efficiency (mean time per epoch on a GPU, over 5 epochs) for fixed (max, sum, PNA), recurrent (GRU), LCM (Binary-GRU) and regularised LCM (Binary-GRU-Assoc) aggregators. The shaded region is bounded above and below by the maximum and minimum values across all runs.

# H   Detailed results and discussion for the PNA synthetic benchmark

**Out-of-distribution performance.**   We present the *out-of-distribution* performance of our aggregators in Figure 5. Note that the MPNN (max) curve corresponds to the second-best aggregator tested out-of-distribution in [Corso et al., 2020], after PNA – this curve stops at graphs of sizes between 45 and 50 as this is the maximum graph size on which the aggregator was tested in the paper.

Observe that all learnable aggregators generalise as well as, or better than, the max-aggregator. Notably, while the Binary-GRU-Assoc aggregator underperforms in-distribution compared to Binary-GRU, it beats Binary-GRU out-of-distribution and performs competitively with GRU: indeed, the *regularisation towards associativity* has improved performance out-of-distribution at the cost of a slight decrease in performance in-distribution.

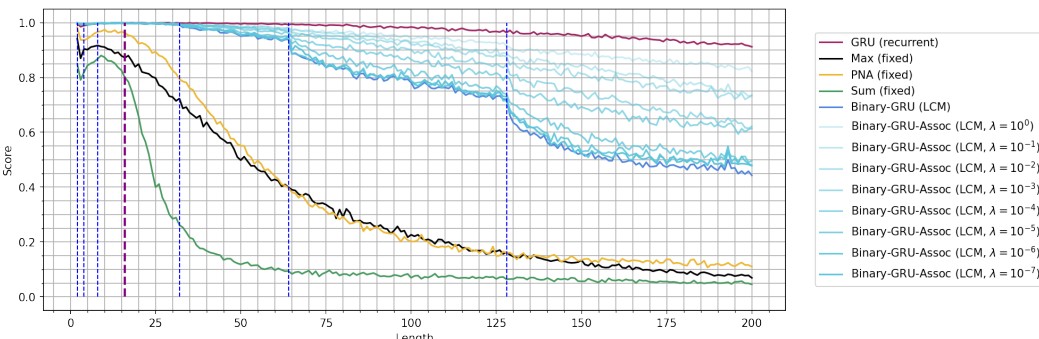

**Figure 4:** Mean generalisation performance for fixed, recurrent and LCM aggregators, sweeping across regularisation rate $\lambda$ for Binary-GRU-Assoc.

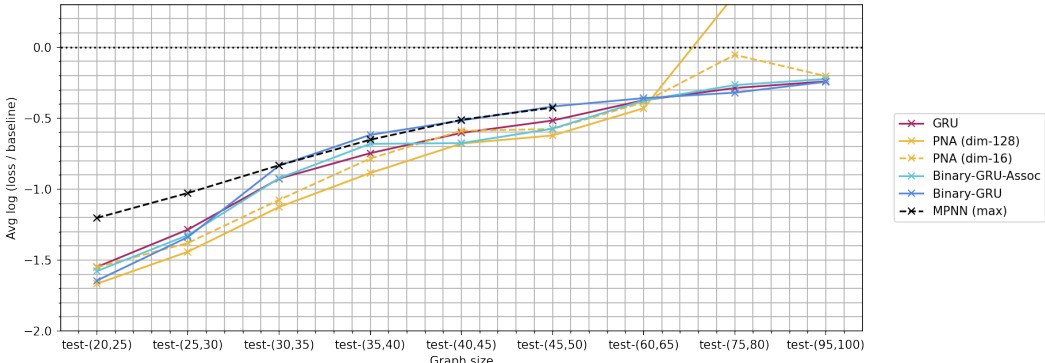

**Figure 5:** Mean generalisation performance (multi-task $\log_{10}$ of the ratio between the MSE loss for the GNN and the MSE loss for the baseline) for fixed, recurrent and LCM aggregators on the PNA multi-task benchmark.

Notice also that all learnable aggregators are *more stable* than PNA for very large graphs – in fact, the 128-dimensional PNA *explodes* for graph sizes above 75.

**Dimensionality and overfitting.**    Finally, we take a look at the *effects of high dimensionality* on the performance of various aggregators.

For **learnable aggregators**, increasing dimensionality seems to help performance. We demonstrated that, if learnable aggregators operate over a latent space with a high enough dimension, they can beat individual fixed aggregators on tasks the fixed aggregators should be aligned to, and can even be competitive with PNA. Informal testing showed that the performance of learnable aggregators drops substantially if the dimensionality of these aggregators is reduced.

By contrast, for **fixed aggregators**, increasing dimensionality seems to harm performance: Corso et al. [2020] found that "even when [models with fixed aggregators] are given 30% more parameters than the [model using] PNA, they are qualitatively less capable of capturing the graph structure". (And for this reason, we did not test models with fixed aggregators in the 128-dimensional setting.)

For **PNA**, the story is slightly more complex: while the 16-dimensional PNA performs well in-distribution (and, to some extent, out-of-distribution), this improvement in performance is small, especially when compared to PNA's standard deviation. And notably, unlike the 16-dimensional PNA, the 128-dimensional PNA *explodes* out-of-distribution.

So it seems that, *when increasing the dimensionality of the aggregator, fixed aggregators may have more of a tendency to overfit*.

One possible hypothesis for this phenomenon comes from observing that, by Section 2.2,

- in cases where the problem we're attempting to solve aligns with the fixed aggregator we want to use, we can often learn a simple homomorphism from the fixed aggregator to our latent space, and

- while homomorphisms from fixed aggregators are expressive enough in principle to model any commutative monoid, the required homomorphism is complex and doesn't generalise out-of-distribution.

Note that, even for choices of fixed aggregator where some tasks align with the underlying monoid, the aggregator still doesn't align perfectly with the **combined 'multitask benchmark monoid'** that we would need to learn to imitate in order to perform all tasks simultaneously. So, if we have the dimensionality to do so, our fixed aggregator may try to combine the existing monoids to approximate this multitask monoid in-distribution, in a way that does not generalise. In other words, it may be easier to get better performance by learning a very complex homomorphism from our fixed aggregator that works well in-distribution but struggles to extrapolate, than by learning a simple homomorphism from the fixed aggregator that 'mostly works'.

Under this hypothesis, *low-dimensional feature spaces provide an inductive bias towards learning simple homomorphisms that generalise out-of-distribution*.

# I   Reference table for code snippets

Throughout this work, we present code snippets in Haskell [Marlow et al., 2010], a statically typed, purely functional programming language.

As the fundamental idea behind this work – using *algebraic structures* as a means of abstraction in software development – was popularised by Haskell and its surrounding community, we observe that the ideas presented in this paper are most concisely stated through the lens of Haskell.

Furthermore, in the spirit of Olah [2015], we observe that there is a very close correspondence between the construction of neural networks and the construction of purely functional programs: indeed, we believe that strongly typed, purely functional languages like Haskell offer great potential for safe, succinct specification and training of neural networks.

For those unfamiliar with Haskell, we present the Haskell snippets featured in the main body of this work, alongside roughly equivalent implementations in Python.

| Haskell | Python |
|---|---|

```haskell
class CommutativeMonoid a =
  e :: a
  <> :: a -> a -> a

{-
where commutative monoids M satisfy
  x <> e == e
  x <> y == y <> x
  x <> (y <> z) == (x <> y) <> z
-}
```

```python
class CommutativeMonoid(Protocol, Generic[A]):
    @staticmethod
    def id() -> A:
        ...

    @staticmethod
    def plus(a: A, b: A) -> A:
        ...

    @classmethod
    def reduce(cls, xs: List[A]) -> A:
        accumulator = cls.id()
        for x in xs:
            accumulator = cls.plus(accumulator, x)
        return accumulator

"""
where commutative monoids M satisfy
  M.plus(x, M.id()) == x
  M.plus(x, y) == M.plus(y, x)
  M.plus(x, M.plus(y, z)) == M.plus(M.plus(x, y), z)
"""
```

**Snippet 1:** *Defining the interface for commutative monoids.* In Haskell, we do this by specifying a typeclass, such that a commutative monoid over some type T is defined by giving an instance of the typeclass for type T. In Python, we do this by defining an abstract class, such that a commutative monoid over some type T is defined by specifying a child class of CommutativeMonoid[T].

| Haskell | Python |
|---|---|

```haskell
type M = (Int, Int)
instance CommutativeMonoid M where
  e = (infinity, infinity)
  (a1, a2) <> (b1, b2) = (c1, c2)
    where c1:c2:_ =
      sort [a1, a2, b1, b2]

secondMinimum :: [Int] -> Int
secondMinimum = dec . agg . map enc
  where
    enc x = (x, infinity)
    agg = reduce (<>)
    dec (_, x2) = x2
```

```python
class SecondMinCM(CommutativeMonoid[Tuple[int, int]]):
    @staticmethod
    def plus(
        a: Tuple[int, int], b: Tuple[int, int]
    ) -> Tuple[int, int]:
        c1, c2 = sorted([*a, *b])[:2]
        return (c1, c2)

    @staticmethod
    def id() -> Tuple[int, int]:
        return (INFINITY, INFINITY)

def secondMinimum(xs: List[int]) -> int:
    encoded = [(x, INFINITY) for x in xs]
    (_, x2) = SecondMinimumCM.reduce(encoded)
    return x2
```

**Snippet 2:** *Defining the 2nd-minimum commutative monoid.*

| Haskell | Python |
|---|---|

```haskell
fold :: (a -> b -> b) -> b -> [a] -> b
fold f z [] = z
fold f z (x:xs) = f x (fold f z xs)
```

```python
def fold(f: Callable[[B, A], B], z: B, xs: List[A]):
    accumulator = z
    for x in xs:
        accumulator = f(accumulator, x)
    return accumulator
```

**Snippet 3:** *Implementing a polymorphic fold over lists.* Note that, for idiomatic reasons, the Haskell implementation presents a *right fold*, whereas the Python implementation presents a *left fold* – i.e. when folding $f$ over a list $[a, b, c]$, the Haskell implementation would return $f(a, f(b, f(c, z)))$ whereas the Python implementation would return $f(f(f(z, a), b), c)$.

| Haskell | Python |
|---|---|

```haskell
rnnCell :: Learnable
  (Vec R h1 -> Vec R h2 -> Vec R h2)
initialState :: Learnable (Vec R h2)

rnn :: Learnable
  ([Vec R h1] -> Vec R h2)
rnn = fold rnnCell initialState
```

```python
rnnCell: Callable[
    [HiddenState, InputState], HiddenState
]
initialState: HiddenState

def rnn(inputs: List[InputState]) -> HiddenState:
    return fold(rnnCell, initialState, inputs)
```

**Snippet 4:** *Implementing an RNN as a fold over lists.* Note that, as mentioned in Snippet 3, the RNN as implemented in Haskell will consume its list of input states 'in reverse'.

| Haskell | Python |
|---|---|

```haskell
binOp :: Learnable
  (Vec R h -> Vec R h -> Vec R h)
identity :: Learnable (Vec R h)

type HiddenState = Vec R h
instance (CommutativeMonoid
  HiddenState) where
    e = identity; <> = binOp

aggregate :: Learnable
  ([HiddenState] -> HiddenState)
aggregate = reduce (<>)
```

```python
binOp: Callable[[HiddenState, HiddenState], HiddenState]
identity: HiddenState

class LearnableCommutativeMonoid(
    CommutativeMonoid[HiddenState]
):
    @staticmethod
    def plus(
        a: HiddenState, b: HiddenState
    ) -> HiddenState:
        return binOp(a, b)

    @staticmethod
    def id() -> HiddenState:
        return identity

def aggregate(xs: List[HiddenState]) -> HiddenState:
    return LearnableCommutativeMonoid.reduce(xs)
```

**Snippet 5:** *Defining a learnable commutative monoid over hidden states.* We assume we have access to a learnable binary operation $binOp \in (\mathbb{R}^h \times \mathbb{R}^h) \to \mathbb{R}^h$ and a learnable vector $identity \in \mathbb{R}^h$.

| Haskell | Python |
|---|---|

```haskell
binaryGRU :: Learnable
  (Vec R h -> Vec R h -> Vec R h)
binaryGRU v1 v2 = do
  g <- new (gruCell
    (InputDim h) (HiddenDim h))
  return (g v1 v2 + g v2 v1) / 2
```

```python
class BinaryGRU:
    def __init__(self, h):
        self.gruCell: Callable[
            [HiddenState, HiddenState], HiddenState
        ] = GRUCell()

    def __call__(self, x: HiddenState, y: HiddenState):
        return (
            (self.gruCell(x, y) + self.gruCell(y, x)) / 2
        )
```

**Snippet 6:** *Defining the Binary-GRU operator.* In Haskell, we present this via a (hypothetical) monadic API for defining neural networks; in Python, we define a class in the style of TensorFlow / PyTorch modules.

