# OpenReview forum: "Learnable Commutative Monoids for Graph Neural Networks"
_logconference.io/LOG/2022/Conference — LoG 2022 Poster_

### Official Review · Reviewer_eAXz · 2022-10-13

**Overall Score:** 6
**Confidence:** 4

**Review:**

### Summary
In this paper, the authors define adaptive commutative monoids for graph neural networks. The paper is inspired by the observation that the sum over neighbor nodes in a graph over discrete inputs can approximate arbitrary permutation invariant functions but fails for many graph domain problems. Recently, recurrent neural networks have been proposed in the literature as a solution that has a depth of O(V) (V for the set of vertices), which poses a particular difficulty for parallelization and training on large graphs. Inspired by the fact that a well-behaved aggregator for a GNN forms a commutative monoid over the latent variable space, the authors propose a framework for constructing learnable commutative monoids with aggregators of depth O(log V) that have similar performance to recurrent aggregators, but an exponential improvement in parallelizability and length dependence. These claims are demonstrated with the help of empirical experiments.

### Review
First, I consider the main points that I find critical about this work. The advantages mentioned in the abstract due to the exponential improvement in terms of paralellisability as well as the dependence on the depth of the graphs were not picked up explicitly by the experiments. However, this would be extremely useful as these are core arguments for the whole paper. Also, it is not made clear to the reader why a classification problem, just one of many learnable problems, is particularly suitable for demonstrating the underlying theoretical structures. The emphasis here is on comparability with different aggregators rather than on finding experiments that might present interesting limiting cases of commutative monoids. It is also not entirely clear to the reader whether the theorems proved in this case are valid for two monoids with different cardinality. Here an explanation on the part of the authors would be very desirable or a corresponding remark on the finite case within the paper.
I would then like to highlight which parts of the paper are a valuable contribution to the Learning on Graphs conference. First of all, the problem is mathematically well-defined at the beginning and the corresponding assertions about the structures of homomorphisms between commutative monoids are proved in the proposed framework. Further, throughout the paper, every intermediate result and assertion has been very well incorporated or verified in the existing literature. The experiments are presented very clearly. In total, ten different models were tried with synthetic datasets as well as datasets from computer vision (MNIST and CIFAR). No further justification was given for the choice of data sets. The usual Benchmarks for graph neural networks were used, which makes the work very comparable to other studies. Overall, the learnability of commutative monoids is very clearly formulated. The authors also manage a certain balancing act between notation from pure mathematics and theoretical computer science. Moreover, the experiments in Sect. 3.3.1 are described in detail and have been observed thoroughly. A much more detailed description of the results, which is not inferior in clarity to the core of the work, can be found in Appendix G. The implementation details for Haskell can be found in Appendix E and appear to be correct. The work is successful in that learnable commutative monoids have been formally defined and experimentally implemented, and sufficient documentation and code material has been provided to validate the stated results.

### Strong points
- Noticeably clear description of the structure of the work in the abstract and the expected results. The main point is sold well, and the style is also spot on.
- The required arguments and claims are supported throughout the paper with novel references and corresponding theorems from the literature. Furthermore, the placement of the problem and the own solutions within the literature is very well-conducted (see L. 81-84).
- The structure of the work is clear and traditional. The contents are clearly communicated and there is no unnecessary information. All propositions are proven in the appendices. The presentation of the experimental results is well done, easy to follow and easy to read.
- The problem to be solved is beautifully motivated in Sect. 2 by an explicit definition of the actual problem with reference to existing literature (see L. 62-67).
- Sect. 3: For each experiment carried out, well-founded expectations are always stated without giving away the results in advance, which will be given in full in the later part of the work. This is stylistically highly successful and encourages an interest in reading more.
- The representations of all diagrams and experiments are chosen very beautifully, illustratively and sparingly in space. Overall, the writing of the work is very well done.
- Sect. 3.2.2.: State-of-the-Art Benchmarks: connectedness, diameter, spectral radius, shortest path, eccentricity and Laplacian of the graph.

### Weak points
#### Formalities
- In this paper, italicisation is used both to emphasise technical words and for intentional emphasis within the semantic structure of the sentence, as well as for object language. I would recommend using italicisation exclusively for object language, as is common in most sciences, and not italicising anything else. What is also possible is to commit to a single semantic meaning of italicisation and use it consistently throughout the paper.
- L. 4-6: At this point, reference is made to recently published works. You are also welcome to cite the relevant reference in the abstract. As a reader, I would have been very interested in it.
- L. 53-54: "[...] depth linear in n [...]" should be something such as "[...] with a depth that is linear in n [...]".
- L. 101: "Suppose this GNN learned to imitate the M-aggregation commutative monoid - in other words, that there exists a function [...]" should be changed to " Suppose there exists a function [...], in other word this GNN learned to imitate the M-aggregation commutative monoid.". Please always prefer the formally correct description of the mathematical property.
- L. 105: A dot is missing at the end of the line. Punctuation marks should also be included in equations.
- L. 112-113: "[...] in other words, if the homomorphism doesn't have to do much work [...]", this is simply too colloquial.
- L. 129-130: The term in-distribution is also written as "in distribution". Please standardise.
- Footnote 1: "where <g(M)> denotes the submonoid of F generated by g(M)" should be changed into "<g(M)> denotes the submonoid of F generated by g(M).". Please note the punctuation mark at the end of the line.
- L. 145: I would recommend a double colon instead of a full stop as a punctuation mark.
- Footnote 5: Blank space before the full stop.
- L. 217: Format and line spacing should be corrected.
- Figure 1: The diagram is very well presented, but the font is a little small. My recommendation would be to place the legend within the diagram, e.g. semi-transparent, and thus enlarge the diagram itself, likewise the labelling of the axes and the labelling of the axis sections could then be enlarged.
- L. 477: "[...] for A, B multisets of M." should be changed into "[...] for some multisets A,B of M.".
- L. 478, 484: "and we're done", I recommend removing this as it is too colloquial.
- L. 480: "[...], have that h(F) [...]" into "[...], we have that h(F) [...]".
- L. 560: Here, a punctuation mark is also missing at the end of the enumeration, once a comma and once a full stop, because enumerations do not cancel out the punctuation rules.

#### Contents
- L. 9: Aggregators invariant to what? I suspect to permutation.
- L. 18: It would have been nice to provide the proposed LCM (learnable commutative monoid) aggregator with theoretical guarantees in addition to the empirical experiments. As a reader, I do not know what is meant by "best of both worlds".
- L. 28: Please specify what is meant by simple at this point.
- L. 30: I believe that downstream-task is a term used in computer science, but less technical and more colloquially illustrative. I would speak of learning problem at this point.
- L. 30: Throughout the paper, the term align is used for aggregators. I do not understand what it technically means. Is it about falling below an error under an optimisation by certain distance functions? That should be explained.
- L. 41: Here I was a bit thrown for a loop as to between which structures the homomorphism should map. I would add that we are in the category of commutative monoids, because this only becomes clear when we read on (it is mentioned in the headline of Sect. 2.1). Since no theorems from category theory are used, it is not necessary to speak of a category, but at least of homomorphisms between commutative monoids.
- L. 57: What is this "sweet spot"? I would prefer a less colloquial description of what the LCM can achieve. A mathematical description of this sweet spot would be ideal.
- L. 58: How is the expressivity of RNNs defined?
- L. 88: Complex is used to colloquially, for my taste. Complex in which sense?
- L. 122: What is a natural homomorphism? The problem here is that there are indeed natural transformations that transport structures between functors within two categories. If one describes the homomorphism here as natural, then the semantic reference is very close. Better perhaps to speak of canonical, but then one would have to specify that one does not have to make a choice for such a homomorphism for construction.
- L. 129-130: Please define what in-distribution and out-distribution should mean.
- Sect. 2.4: How does learning work for commutative monoids that do not have the same cardinality? This has relevance for theoretical guarantees. For example, if one considers countable and uncountable monoids, one would always have an absorbing element for a homomorphism. I also don't see how Proposition 1 can be readily true if the multiset X is chosen to be not finite but infinite, or uncountable infinite. But the proof in the appendix seems to assume a finite multiset, since the operation over the entire set is equated with the operation over a certain finite index of the indexed elements of this set. Would it be possible to comment on this and explicitly write that any cases are covered by preposition 1 (with proof) or to restrict X as a finite multiset? (Appendix A: The question of cardinality came to me when, without further ado, the operation over the entire set was equated with the operation over a finitely induced subset in Proposition 1 at the back direction.)
- L. 245: What is meant by regularisation "towards algebraic axioms"? This should be explained in more detail.

### Recommendation
Overall, the work is theoretically sound and experimentally validated. The work has been very well placed in the existing literature. All claims are always either experimentally proven, demonstrated, or supported by references to papers in the relevant literature. The writing style of the paper is a delight. The structure is classically kept. I recommend a thorough review for minor spelling errors, but these do not carry much weight in the overall rating. Since I believe that the implementation details are extremely useful and sufficiently detailed for the Learning on Graphs conference community and consideration of adaptive commutative monoids in graph neural networks can be considered work belonging to the canon, I recommend that the submitted work be accepted for publication with reservations regarding the open issues I raised.

---

### Official Review · Reviewer_7QLX · 2022-10-16

**Overall Score:** 6
**Confidence:** 3

**Review:**

Summary: This paper proposes a learnable aggregator for graph neural networks. The implementation is based on a symmetry GRU, combined with a loss to encourage the commutative monoid. The experiments demonstrate the effectiveness of the proposed method.

Strong points:
1. This paper makes a good summary of the aggregator in graph neural networks.
2. The proposed method seems simple and effective.

weak points:
1. Some presentations are not clear, for example, the arrangement of "\paragraph"s in Sec 2.5 looks messy. Some terms in the algorithms based on functional programming are not clear. Is it possible to provide a reference notation table?
2. The proposed method cannot beat the previous baseline "GRU" in performance.

questions:
1. Have authors tried other learnable aggregator backbones instead of GRU?
2. What do you think about the future of the "learnable aggregator"? Will it be replaced by better edge features or message passing with a fixed and simple aggregator?

---

### Official Review · Reviewer_yX5D · 2022-10-21

**Overall Score:** 6
**Confidence:** 2

**Review:**

**Summary:** This paper comprehensively compares various aggregation functions for graph neural networks. Specifically, the authors analyze the limitations of fixed and recurrent aggregators which motivates the proposed learnable commutative monoid (LCM) aggregators. The proposed method is theoretically supported and empirically demonstrated to be effective in three different tasks using synthetic and real-world datasets.

**Strong points:** **(1)** Considered methods are theoretically analyzed. **(2)** The proposed method theoretically and empirically addresses previous methods. **(3)** Experimental results are well-analyzed. Specifically, the authors analyzed several different aspects (e.g., in-distributions, per-task).

**Weak points:** **(1)** The authors conducted experiments on three different tasks which are quite intuitive. Can the proposed method be directly used to solve classical graph-related downstream tasks, such as node classification and link prediction?  **(2)** Pseudo-codes provided by the authors are non-intuitive to understand. Can the codes be rephrased into more popular languages (e.g., Python)?

From the above strong/weak points, I recommend the paper as a *weak accept*.

---

### Official Review · Reviewer_E3TN · 2022-10-21

**Overall Score:** 6
**Confidence:** 3

**Review:**

Summary:

This work is focused on aggregation functions in GNNs. It builds a general framework of aggregations functions based on commutative monoid, within which several categories of functions are included. It discusses disadvantages of popular fixed or recurrent aggregators. Then it proposes a general method to build a (perhaps approximately) commutative and associative aggregator from any binary operator. Sufficiency and efficiency of the proposed method is verified with synthetic and real experiments.

===

Pros:
1. The aggregator is a very fundamental topic in the GNN domain. And this work is also trying to tackle problems in fundamental way, via building a general framework containing MPNNs with any aggregator, which makes the discussion clean to improve the aggregator itself.
2. The discussion on fixed or recurrent aggregator is comprehensive supported by sufficient experiments. The proposed method enjoys clear conceptual and numerical advantages against baselines.
3. While sequence-based aggregator is not novel for GNNs, this work finds a great point to improve it, with the motivation of strengthening its efficiency (reducing the number of recurrent layers). For this, it naturally introduces associativity, which gives us binary trees, reducing the depth from linear to log.

Cons:
1. The main flaw is its presentation including many Haskell expressions. Although it delivers messages efficiently and accurately, it may be not easy for general audience to follow. This might be due to authors' background.

===

Questions:
1. Since GRU itself is not commutative or associative, I am wondering what would happen if a simple MLP is used as the base operator. Although it is not binary, we can treat it as permutation-sensitive encoder, and perhaps clipping and padding the input makes it compatible with input in various lengths. This sounds like Janossy pooling or Relational pooling from Murphy et al. The advantage of MLP is better efficiency than GRU.

===

Suggestions:
1. It would be great if definitions and algorithms can be expressed in mathematical forms or pseudo-code.

===

Typo:
1. Line 99: Appendix 4 -> Appendix A

===

Final comment:

For now I recommend a weak acceptance due to the presentation issue as mentioned above. I am willing to raise my score if it can be resolved.

---

### Meta-Review · Area_Chair_f1Vf · 2022-11-17

**Confidence:** 4
**Recommendation:** Accept

**Meta Review:**

The reviews are quite unanimous in their view of the paper: it is sound and gives a unique and valuable perspective on learnable aggregation functions. Several reviewers raised the point that Haskell expression are not native to many of the intended audience; this issue has been addressed by the authors by also providing python snippets. Although the idea of learnable aggregators is not novel, the abstract viewpoint taken in this paper, and its embedding relative to literature make it an important contribution. The work does contain novelty in the improving RNN based aggregators in terms of efficiency, and supporting the concept with experiments. Several reviewers have raised clarifying questions which have been addressed and incorporated by the authors.

Given the overall soundness and quality of the paper I recommend accept.

---

### Decision · Program_Chairs · 2022-11-23

Accept (Poster)